# WWW.Serve: A Decentralized Framework for Collaborative LLM Serving

## Abstract

Large language model (LLM) services are mostly centralized, causing inherent scalability bottlenecks and leaving substantial scattered GPU resources underutilized. Decentralized serving could potentially address these limitations, but impose challenges of **trust**, as the identity and behavior of participants cannot be reliably regularized, and **fairness**, i.e., how to maximize the benefit of all resource providers to improve engagement. However, existing decentralized frameworks **predominantly emphasize the rights and protections of users and the cooperative aspect among GPU providers** while **overlooking the inherent competitive dynamics**, imposing substantial constraints on GPU providers, such as requiring them to accept excessive platform-level oversight and to execute all assigned requests with fixed software stacks on fixed hardware configurations. We argue that such assumptions are unrealistic in real-world decentralized environments. To this end, we propose **WWW.Serve**, a decentralized framework for interconnecting LLM service worldwide. It preserves the flexibility of service providers, allowing them to decide **when, under what policies, and with what resources** they join the decentralized network, while further ensuring their anonymity. In terms of efficiency, WWW.Serve supports self-organizing request dispatch, enabling the network to autonomously allocate requests without centralized coordination. Three key designs are integrated: a blockchain-inspired credit system for trustless collaboration, gossip-driven peer synchronization for flexible participation, and a duel-and-judge mechanism for robust contributor evaluation. Empirically, we show that WWW.Serve incentivizes higher-quality services to obtain greater profit, while improving global SLO (service-level-objective) attainment by up to $1.5\times$ and lowers latency by $27.6\%$. Its performance approaches, and in some cases surpasses, centralized scheduling, while fully preserving the benefits of decentralization. These results highlight WWW.Serve as a promising foundation for real-world, decentralized LLM serving.

## 1 Introduction

Large language model (LLM) are becoming popular. With increasing deployments of LLM service and prices of GPU, distributed LLM serving has become essential for mitigating workload fluctuations and leveraging potentially idle hardware resources. Centralized scheduling (Zheng et al., 2024; Kwon et al., 2023), however, constrains the engagement of different entities. Therefore, decentralization has long been recognized as an effective paradigm (Liu et al., 2024; Dong et al., 2025). By relying on peer-to-peer communication (Kermarrec & Taïani, 2015), it improves scalability, adapts to dynamic participation, enhances robustness by eliminating single points of failure, and improves anonymity and privacy (Li & Palanisamy, 2019; Ma et al., 2024).

Despite these apparent advantages, existing decentralized serving systems remain largely impractical in real-world settings: (1) Fundamentally, they **predominantly emphasize the rights and protections of users and the cooperative aspect among GPU providers** while **overlooking the inherent competitive dynamics**, namely, that GPU providers, as the holders of the actual computational assets, are naturally incentivized to maximize their own profit. Existing frameworks (Fang et al., 2025) attempt to rely on a small central organization to impose substantial constraints on GPU providers, such as requiring them to accept excessive platform-level oversight (Fang et al., 2025; Wu et al., 2025) and to execute all assigned requests with fixed software stacks (Mei et al., 2025a;

Borzunov et al., 2023; Mei et al., 2025b) on fixed hardware configurations. Although this may theoretically enable better resource allocation, the regulator itself is untrusted, rendering the approach unrealistic in practice. (2) Besides, providers typically maintain their own prioritized workloads and may experience fluctuations in available resources. This highlights enabling flexible, customizable mechanisms for providers to determine how they engage with the decentralized system.

Ideally, we desire a decentralized framework that acts like an open, competitive market, allowing providers to decide **when, under what policies, and with what resources** they join the decentralized network. At the same time, such a framework should: **1.** provide a well-designed reward mechanism that incentivizes providers to deliver higher-quality services, including faster hardware, more user-oriented scheduling policies, better serving systems, and higher-quality models. Such incentives should further encourage innovation (e.g., in models, systems, or kernels), enabling providers to offer superior services at lower cost. **2.** enable market-driven exchange of computational capacity, where overloaded nodes can outsource requests while underutilized nodes capitalize on idle resources, allowing compute supply and demand to self-balance through decentralized interactions. **3.** incorporate a principled routing protocol to improve global efficiency under highly dynamic and unpredictable resource availability. However, to meet these demands, three fundamental questions arise. In the following, we discuss these challenges and outline our key approaches to address them.

**Question 1.** How can the system enable trustworthy market-driven trade of computational capacity, i.e., implement reliable request scheduling among anonymous participants without central coordinators? Achieving this requires a way to quantify each participant's contributed capacity and use it to guide task allocation. To this end, we introduce a credit-based transaction system that functions as a reputation-like indicator under anonymity: participants earn credits by serving delegated requests and spend them when offloading their own tasks. Request routing is then guided via a Proof-of-Stake-based (PoS) mechanism, in which participants' staked credits, freely adjust according to their own strategy, determine their likelihood of being selected to execute delegated requests. This design allows high-load servers to offload tasks to relieve pressure and improve user satisfaction, while low-load servers utilize idle resources to earn credits for future offloading. By accumulating credits through contribution, participants effectively engage in a decentralized market for computing power.

**Question 2.** How can we incentivize participants to provide high-quality services, thereby improving overall user experience? In an anonymous network, providers naturally seek to maximize their own gain. This competitive dynamics, however, creates the risk that participants may deploy low-quality services to "exploit" the contributions of others, undermining overall system performance. To address this, we must align individual incentives with service quality. To this end, we introduce a duel-and-judge mechanism: a subset of requests is collectively evaluated collectively within the network through pairwise comparison, with the superior response receiving a credit reward and the inferior response incurring a penalty. This design enables dynamic credit redistribution based on service quality. When combined with PoS-based request scheduling, it can be proved that low-quality nodes are gradually phased out of active participation, reinforcing the network's overall service quality and fostering decentralized incentives for correctness.

**Question 3.** How can the system remain robust under highly dynamic and unpredictable resource availability? In real-world scenarios, individual infrastructures may suffer from hardware failures, network disconnections, or user-driven constraints, all of which lead to unstable participation of resources. To address this challenge, we design a lightweight gossip-driven protocol that enables dynamic online and offline participation. Each participant periodically exchanges availability information with a subset of peers and reconcile discrepancies. Through this protocol, newly joined resources can be quickly integrated into the network, while sudden departures or failures can be rapidly detected. Without relying on central coordinators, lightweight pairwise exchanges allow information updates to diffuse across the network and converge quickly, ensuring stable and reliable service despite the volatility of global-scale resources.

Having addressed these challenges, we introduce **WWW.Serve**, a decentralized framework for collaborative LLM serving. In general, our main contributions are:

- We present **WWW.Serve**, a fully decentralized system that operates as an open, competitive market of computational capacity, enabling request routing and workload balancing among distributed and anonymous LLM servers.

- We design three core mechanisms to ensure reliability: a credit-based transaction system for trustless request delegation, a gossip-driven protocol for dynamic peer synchronization, and a duel-and-judge mechanism for contributor evaluation.
- We provide a game-theoretic analysis proving that our collaborative framework converges to equilibria that sustain high-quality LLM service even under full anonymity.
- Empirical results demonstrate that WWW.Serve achieves near-centralized efficiency, improving global SLO attainment by up to $1.5\times$ and reducing latency by up to $27.6\%$, while sustaining robustness under dynamic participation and supporting flexible collaboration policies.

The rest of this paper is organized as follows. Section 2 reviews related work, Section 3 introduces the architecture of WWW.Serve, and Section 4 details its core mechanisms. Section 5 provides a game-theoretic analysis, Section 6 reports empirical results, and Section 7 concludes.

## 2 RELATED WORK

**Decentralized Computing.** Early volunteer-based platforms (Anderson et al., 2002; Foster & Kesselman, 2003; Anderson, 2019; Shirts & Pande, 2023) demonstrate the feasibility of harnessing distributed resources for large-scale scientific workloads. With the advent of blockchain (Nakamoto, 2008), decentralized frameworks like Ethereum (Song et al., 2024) introduce trustless execution environments where tasks are handled transparently and verifiably through smart contracts. Subsequent systems such as Filecoin (Labs, 2017) and Golem (Network, 2020) extend this model with incentive mechanisms such as Proof-of-Stake (Kiayias et al., 2017; Buterin & Griffith, 2019), ensuring fair contribution and deterring malicious behavior. These systems highlight the importance of incentive alignment and trustless coordination, motivating our decentralized LLM serving design.

**Large Language Model Serving.** LLMs demand substantial computational resources, thus are primarily deployed by service providers such as OpenAI (OpenAI, 2022), Anthropic (Anthropic, 2023), and Microsoft Azure (Microsoft, 2023), offering users online inference services. Meanwhile, the rapid rise of open-sourced, especially reasoning-oriented models such as DeepSeek-R1 (DeepSeek-AI, 2025), LLaMA 3.1 (Touvron et al., 2024), and Qwen3 (Yang et al., 2025) series, enables broader community access and deployment, therefore creating massive demand for high-throughput inference services. In response, a spectrum of LLM serving systems has been proposed.

At the single-model level, SGLang (Zheng et al., 2024) and vLLM (Kwon et al., 2023) leverage various advanced techniques to improve request concurrency and maximize inference efficiency. HexGen (Jiang et al., 2024) and Helix (Mei et al., 2025b) provide adaptive scheduling strategies that optimize model deployment and task migration across heterogeneous resources. Furthermore, DistServe (Zhong et al., 2024) partitions prefill and decoding computations across multiple GPUs, while speculative decoding (Chen et al., 2023; Leviathan et al., 2023; Miao et al., 2024) and sequence-length-aware scheduling (Qiu et al., 2024) offer complementary performance gains. However, these approaches remain inherently centralized and emphasize intra-model performance, without offering systematic solutions for workload balancing across multiple LLM servers.

Recently, decentralized approaches have been further explored, yet they fall short of fully realizing our desired goals. Petals (Borzunov et al., 2023) supports collaborative deployment of a fixed LLM across volunteer GPUs, limiting flexibility in multi-model scenarios and cannot adapt to dynamically changing resources. DeServe (Wu et al., 2025) offers a privacy-preserving offline serving system where users contribute inference capacity collectively, yet still depends on partial centralization for request dispatching and lacks mechanisms to ensure service quality. GenTorrent (Fang et al., 2025) distributes and executes model shards, but relies on trusted organizations to prevent malicious behavior, and therefore does not achieve full decentralization. Other works (Kozgunov et al., 2024; Xian et al., 2024; Chen et al., 2025; Mia & Amini, 2025) explore secure decentralized training and inference frameworks that integrate cryptographic and blockchain-based trust mechanisms. While relevant as background, these approaches do not directly address the specific challenges we target.

## 3 WWW.SERVE'S OVERVIEW

We begin by presenting the overall network architecture of WWW.Serve (Subsection 3.1), followed by a description of the request routing process and node design (Subsection 3.2).

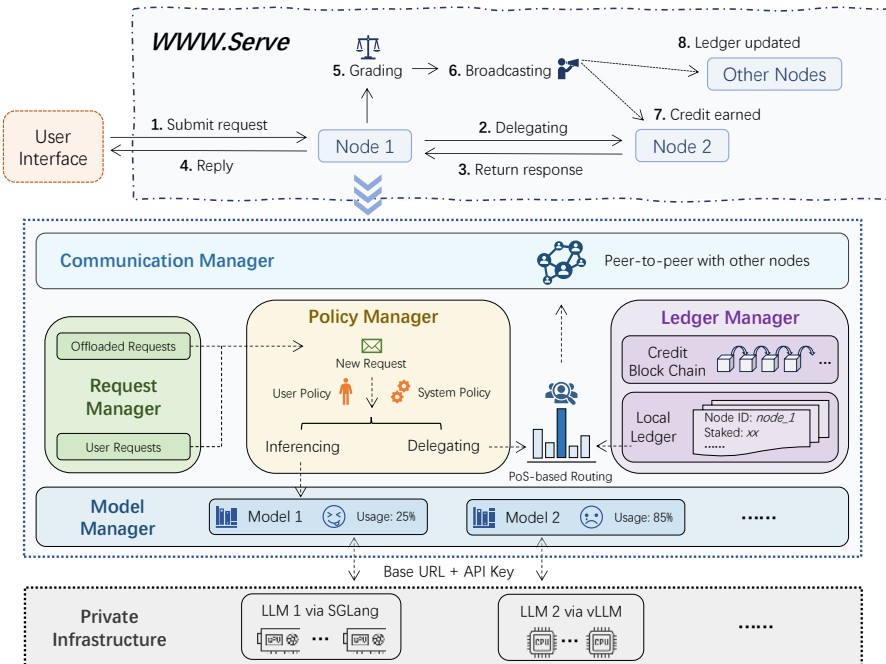

Figure 1: Overview of WWW.Serve. The upper part illustrates the decentralized request routing workflow, while the lower part details the internal architecture of a single node.

## 3.1 NETWORK ARCHITECTURE

As illustrated in Figure 2, WWW.Serve establishes a fully decentralized peer-to-peer network connecting users with LLM service providers.

From the user's perspective, WWW.Serve provides a seamless serving interface. Users do not need to be aware of the underlying decentralized infrastructure; instead, they simply submit inference requests and wait for responses, just as they would with conventional LLM online services. The framework automatically handles request routing, resource discovery, and response evaluation. This design greatly lowers the barrier to adoption, allowing users to access global LLM services without requiring specialized knowledge of network topology or coordination protocols.

From the service provider's perspective, WWW.Serve offers a simple yet flexible participation model. Providers can contribute surplus computational resources without exposing sensitive information, while retaining full control and anonymity within the ecosystem. They are free to join or leave at any time, enabling adaptive scheduling and resource allocation. This design encourages broader participation for service providers, converting idle capacity into valuable contributions for LLM serving.

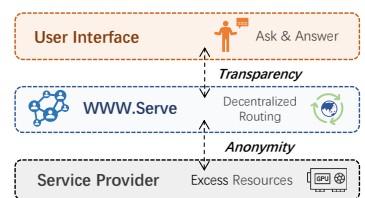

Figure 2: General network architecture.

## 3.2 REQUEST ROUTING AND NODE DESIGN

As illustrated in Figure 1, the inference request in WWW.Serve follows a decentralized routing process that shapes the modular design of each node. This process involves four key stages:

**Request admission.** When a user submits an inference request, it first enters the local request queue maintained by the *Request Manager*, which handles both user-originated and delegated requests. This ensures orderly processing while decoupling admission from execution.

**Scheduling and policy enforcement.** The queued request is then subject to the service provider's configurable policies. The *Policy Manager* decides whether to execute the request locally or delegate

it to other nodes, considering factors such as workload thresholds, willingness to delegate requests, and customized load-balancing rules. This design allows service providers to flexibly participate in collaborative serving while retaining full control over their resources.

**Executor selection and trust establishment.** If the request is delegated, the node selects a reliable executor. To this end, the *Ledger Manager* provides access to peers' stake balances. Candidates are sampled via a Proof-of-Stake–based mechanism, where the probability of selection is proportional to their staked credit. Each candidate is further probed to verify its willingness according to its own policy. Once accepted, the request is forwarded, executed locally by the chosen peer, and the response is returned to the originator. The executor is rewarded through a "credits-for-offloading" transaction, while the duel-and-judge mechanism further evaluates response quality (details in Subsection 4.1 and Subsection 4.2).

**Execution across heterogeneous backends.** For requests served locally, the *Model Manager* provides a unified abstraction layer over diverse serving backends. It executes inference, monitors utilization, and preserves intra-model scheduling efficiency. This ensures that heterogeneous resources can be seamlessly integrated into WWW.Serve.

Together, these stages form a request routing pipeline that ensures policy-driven scheduling, trust-aware executor selection, and efficient execution on heterogeneous LLM servers.

## 4 CORE MECHANISMS

In this section, we introduce three core designs of WWW.Serve: (i) the *Credit-based Transaction System* (Subsection 4.1), which incentivizes and regulates request dispatching; (ii) the *Duel-and-Judge Mechanism* (Subsection 4.2), which ensures reliable and trustworthy contributor evaluation; and (iii) the *Policy Framework* (Subsection 4.3), which supports flexible policies for collaboration.

### 4.1 CREDIT-BASED TRANSACTION SYSTEM

Drawing inspiration from real-world transactions, where users pay for premium LLM services (e.g., API token prices), we design a *Credit-based Transaction System* in which each node's computational resources are represented as transferable credits. These serve as a reputation-like measure that enables dynamical workload exchange while providing economic incentives for active and high-quality participation. Beyond the system itself, credits can be anchored to real-world currency, enabling direct monetization of computational contributions and paving the way for practical deployment of WWW.Serve in commercial large-scale inference services.

However, traditional transaction mechanisms are not sufficient in decentralized settings. Without a shared, tamper-resistant ledger, nodes can misreport their actions or selectively reveal inconsistent transaction histories to different peers (Nakamoto, 2008; Cachin & Vukolić, 2017; Bano et al., 2017; Tripathi et al., 2023). For example, a node might claim the same credits have been spent in multiple transactions (double spending), or refuse to acknowledge deductions from failed or malicious executions. Since no single entity holds the authoritative record, such inconsistencies can hardly be reconciled, undermining both fairness and trust across the network.

Table 1: Structure of a Credit Block

| Field | Description |
|---|---|
| Block ID | Hash of the current block |
| Parent ID | Hash of the previous block |
| Timestamp | Time of block creation |
| Operations | List of credit-related records |
| Proposer | Node proposing the block |
| Signature | Digital signature |

To address this, WWW.Serve adopts a blockchain-inspired ledger. Each node maintains a local *Credit Block Chain* that records activities such as staking and rewarding in tamper-resistant blocks (Table 1). Blocks are cryptographically linked, so any modification is immediately detectable. A credit transaction occurs whenever a delegated request is completed. The responsible node records this by creating a new block and broadcasting it to its peers, which independently validate the block. The transaction is finalized once a majority of peers confirm and append it to their local ledgers.

The security of this design relies on two complementary features. First, nodes must stake credits to participate in scheduling, which discourages malicious behavior by putting dishonest nodes' stakes at risk. Second, decentralized verification ensures that every block is independently validated by

multiple peers before being appended to the chain, preventing any single node from manipulating the ledger. Thus, balances are guaranteed to be secure, auditable, and tamper-resistant, all without relying on a centralized authority.

## 4.2 DUEL-AND-JUDGE MECHANISM

In our decentralized serving network, participants are anonymous and heterogeneous, with no central authority to verify the quality of their contributions. This raises a fundamental risk: low-quality or even malicious nodes may provide incorrect results, degrading overall service reliability. Prior frameworks (Bouchiha et al., 2024; Zhang et al., 2024; Fang et al., 2025) rely on verification committees or light evaluation models, but they introduce complexity and privileged roles that limit true decentralization. In response, WWW.Serve introduces the *duel-and-judge mechanism*, enabling peer-driven evaluation of the service quality.

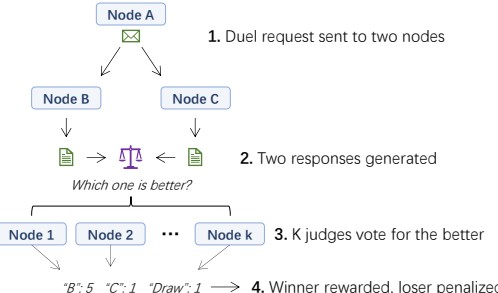

Figure 3: Duel-and-judge mechanism.

As shown in Figure 3, a small fraction of delegated requests are randomly designated as *duel requests* and dispatched to two executors sampled via our Proof-of-Stake–based selection mechanism. Next, $k$ judges (also selected via PoS) perform pairwise comparisons of the responses. The inferior executor is penalized by losing part of its stake, while the superior executor and responsible judges earn additional credits. The results of each duel are broadcast and recorded in the credit ledger, ensuring transparency and accountability.

Such duel-and-judge mechanism offers several key advantages for ensuring reliable and high-quality decentralized serving. First, it leverages a pairwise comparison rather than relying on absolute scores. Prior studies (Zheng et al., 2023; Chiang et al., 2024; Watts et al., 2024) demonstrate that pairwise evaluation of LLM outputs yields higher inter-rater agreement and greater robustness, making it a more reliable way to distinguish between competing responses. Second, the involvement of PoS-sampled judge nodes introduces additional impartiality, mitigating risks of collusion and fostering fairness in the evaluation process. Third, the credit redistribution scheme provides strong economic incentives, aligning node behavior with system reliability and thus driving the network toward high-quality operation. A theoretic analysis of the quality evolution is provided in Section 5.

## 4.3 POLICY FRAMEWORK

WWW.Serve introduces a policy framework that governs both individual node decisions and collective network behavior, which operates along two complementary dimensions:

**User-Level Policies:** enable service providers to manage their resources according to individual objectives. First, each node can freely determine its stake amount, which directly influences its probability of being selected as an executor under the Proof-of-Stake–based scheduling mechanism. This design encourages providers to calibrate their credit commitment according to their willingness and capacity to contribute. Second, nodes may define fine-grained operational conditions for offloading, accepting, or queuing requests at their local backends. For example, one may choose to offload tasks once its local workload surpasses a predefined threshold, to accept external requests only when spare GPU capacity is available, or to prioritize its own user-submitted jobs over delegated ones. Such flexibility not only accommodates heterogeneous resource profiles and business goals, but also fosters a competitive yet cooperative ecosystem where service providers optimize their participation strategies while maintaining overall system efficiency.

**System-Level Policies:** serve as global safeguards to preserve fairness and reliability within WWW.Serve, including mechanisms such as Proof-of-Stake–based routing, the credit-based transaction system, gossip-driven peer synchronization, and the duel-and-judge mechanism. These rules provide the necessary trustless foundation, while user-level policies offer flexibility on top of it.

## 5 GAME-THEORETIC ANALYSIS

In this section, we provide a theoremized proof that WWW.Serve converges to a high-quality equilibrium of collaborative LLM services: high-performing nodes accumulate credit over time, whereas low-quality nodes lose exposure and gradually phase out of the system.

**Assumption 1** (Node parameters). *For each node $i \in \{1, \ldots, N\}$, we have:*

- $q_i \in [0, 1]$, *the intrinsic probability that node $i$ produces a high-quality response;*

- $c_i > 0$, *the per-request operational cost of node $i$;*

- $s_i(t) \geq 0$, *the stake of node $i$ at time $t$.*

**Assumption 2** (System parameters). *The system-level constants are:*

- $\lambda > 0$, *the delegated request arrival rate;*

- $R > 0$, *the guaranteed base reward per delegated request;*

- $p_d \in [0, 1]$, *the probability that a delegated request is selected as a duel;*

- $R_{add} > 0$, *the additional reward for winning a duel;*

- $P > 0$, *the penalty for losing a duel.*

**Assumption 3** (PoS selection and duel mechanism). *We write the PoS selection probability of node $i$ and selection-weighted global average quality as*

$$p_i(t) \ = \ \frac{s_i(t)}{\sum_{j=1}^{N} s_j(t)}, \qquad \overline{Q}(t) \ = \ \sum_{i=1}^{N} p_i(t) \, q_i.$$

*To capture the intuition that a higher network average quality $\overline{Q}(t)$ makes it harder for any individual node to stand out, we model the probability that node $i$ wins the duel as*

$$Q_i(t) \ = \ \tfrac{1}{2}\big(1 + q_i - \overline{Q}(t)\big) \in [0, 1].$$

**Assumption 4** (Stake adjustment). *Rational participants adjust their stakes proportionally to realized expected payoffs. Concretely, for some growth constant $\eta > 0$ we assume*

$$\dot{s}_i(t) \ = \ \eta \, \pi_i(t),$$

*where $\pi_i(t)$ denotes node $i$'s expected payoff rate (defined below in Lemma 1).*

**Lemma 1** (Expected node payoff). *Under Assumptions 1–3, the expected payoff of node $i$ from serving a single delegated request is*

$$\Delta_i(t) \ = \ (R - c_i) + p_d\big[Q_i(t) \, R_{add} - (1 - Q_i(t)) \, P\big].$$

*Consequently, the expected payoff rate of node $i$ under delegated request arrival rate $\lambda$ and PoS selection probability $p_i(t)$ is*

$$\pi_i(t) \ = \ \lambda \, p_i(t) \, \Delta_i(t).$$

*Proof.* A single delegated request always yields the base reward $R$ and incurs cost $c_i$, hence the guaranteed net term $(R - c_i)$. With probability $p_d$ the request becomes a duel; conditional on a duel, the expected duel outcome for node $i$ equals $Q_i(t) \, R_{add} - (1 - Q_i(t)) \, P$. Adding these terms gives $\Delta_i(t)$. Multiplying by the delegated request arrival rate arrival rate $\lambda$ and the selection probability $p_i(t)$ yields the stated expression for $\pi_i(t)$. $\square$

**Proposition 1** (Single-node stake-share dynamics). *Under Assumptions 1–4, the stake share of node $i$ evolves according to*

$$\dot{p}_i(t) \ = \ \frac{\eta \, \lambda}{S(t)} \, p_i(t)\big(\Delta_i(t) - \overline{\Delta}(t)\big), \tag{1}$$

*where $S(t) = \sum_j s_j(t)$ is the total stake in the network, and $\overline{\Delta}(t) = \sum_j p_j(t) \Delta_j(t)$ represents the overall average expected payoff.*

*Proof.* Differentiate $p_i(t) = s_i(t)/S(t)$ to obtain

$$\dot{p}_i(t) = \frac{\dot{s}_i(t)S(t) - s_i(t)\dot{S}(t)}{S(t)^2}.$$

By Assumption 4 we have $\dot{s}_i(t) = \eta\pi_i(t) = \eta\lambda p_i(t)\Delta_i(t)$, and summing over $i$ yields

$$\dot{S}(t) = \sum_j \dot{s}_j(t) = \eta\lambda\sum_j p_j(t)\Delta_j(t) = \eta\lambda\,\overline{\Delta}(t).$$

Substituting these into the derivative and simplifying gives equation 1. $\square$

**Proposition 2** (Group-level stake-share dynamics). *Let $\mathbb{H} \subseteq \{1, \ldots, N\}$ be any subset of nodes, and define its group-level stake share*

$$p_H(t) = \sum_{i \in H} p_i(t).$$

*Define the within-group and outside-group average payoffs*

$$\overline{\Delta}_H(t) = \frac{1}{p_H(t)}\sum_{i \in H} p_i(t)\,\Delta_i(t), \qquad \overline{\Delta}_{\neg H}(t) = \frac{1}{1 - p_H(t)}\sum_{j \notin H} p_j(t)\,\Delta_j(t).$$

*Then the group-level stake share evolves according to*

$$\dot{p}_H(t) = \frac{\eta\,\lambda}{S(t)}\,p_H(t)(1 - p_H(t))\big(\overline{\Delta}_H(t) - \overline{\Delta}_{\neg H}(t)\big). \qquad (2)$$

*Proof.* Summing equation 1 over $i \in H$ yields

$$\dot{p}_H(t) = \frac{\eta\lambda}{S(t)}\Big(\sum_{i \in H} p_i(t)\Delta_i(t) - p_H(t)\overline{\Delta}(t)\Big).$$

Write the network average $\overline{\Delta}(t)$ as the convex combination

$$\overline{\Delta}(t) = p_H(t)\overline{\Delta}_H(t) + (1 - p_H(t))\overline{\Delta}_{\neg H}(t).$$

Substituting this into the previous display and simplifying produces equation 2. $\square$

**Theorem 1** (High-quality equilibrium). *Under Assumptions 1–4, the network converges to a high-quality equilibrium, driven by a subset of superior nodes, thereby promoting reliable and high-quality LLM services.*

*Proof.* From Proposition 2, if there exists a subset $\mathbb{H}$ and a time $T$ such that for all $t \geq T$,

$$\overline{\Delta}_H(t) > \overline{\Delta}_{\neg H}(t),$$

then $\dot{p}_H(t) > 0$, hence $p_H(t)$ is strictly increasing for $t \geq T$. Consequently, high-quality nodes progressively accumulate credit while low-quality nodes lose influence, creating incentives for participants to provide superior services and guiding the network toward reliable and high-quality LLM serving. $\square$

## 6 EMPIRICAL EVALUATION

In this section, we evaluate WWW.Serve under diverse configurations and workload scenarios (implementation details are provided in Appendix A):

- In Subsection 6.1, we show that WWW.Serve improves global SLO attainment by up to $1.5\times$ and reduces latency by $27.6\%$ compared to single-node deployment, achieving efficiency close to centralized scheduling.
- In Subsection 6.2, we demonstrate that WWW.Serve handles dynamic participation gracefully, maintaining service continuity as resources join or leave.
- In Subsection 6.3, we confirm that the duel-and-judge mechanism effectively differentiates high-quality contributors from weak or malicious ones, improving network trustworthiness.
- In Subsection 6.4, we present ablation studies on user-level policies, showing that flexible configurations directly influence workload allocation and SLO attainment.

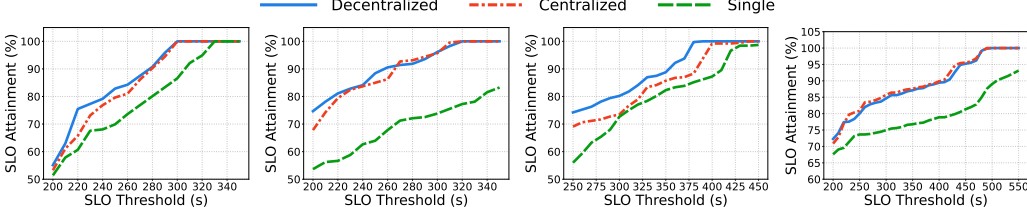

Figure 4: Comparison of global SLO attainment across single-node, centralized, and decentralized (WWW.Serve) deployments under four different experimental settings detailed in Appendix B.

## 6.1 SCHEDULING EFFICIENCY

We first designed a variety of deployment scenarios (details in Appendix B), covering heterogeneous models, diverse GPU hardware, and multiple serving backends. Each node experienced alternating peak and idle periods, simulating realistic fluctuations in service demand. We compared three deployment strategies: single, centralized, and our decentralized scheduling, and measured global Service Level Objective (SLO) attainment (i.e., the proportion of requests completed within predefined latency thresholds) along with the average request latency.

As shown in Figure 4, across all experimental settings, WWW.Serve consistently outperforms single-node deployment and closely matches, in some cases even surpasses, centralized scheduling in terms of SLO attainment. Table 2 further demonstrates that this efficiency translates into substantially lower request latency. Together, these results highlight a key advantage of WWW.Serve: it achieves near-centralized scheduling efficiency without compromising the privacy and autonomy afforded by decentralization.

Table 2: Average request latency comparing different scheduling strategies.

| Setting | Avg. Latency (s) | | |
|---|---|---|---|
| | Single | Centralized | Decentralized |
| Setting 1 | 200.380 | 188.419 | **184.400** |
| Setting 2 | 226.578 | **168.221** | 168.485 |
| Setting 3 | 237.925 | 206.123 | **198.306** |
| Setting 4 | 241.042 | **169.896** | 174.592 |

## 6.2 DYNAMIC PARTICIPATION

WWW.Serve is designed to operate under highly dynamic and unpredictable resource availability in real-world scenarios. We thus evaluate its ability to adapt to arbitrary node arrivals and departures.

The left panel in Figure 5 illustrates nodes joining the network sequentially, starting with two active nodes. When the workload temporarily exceeds available resources, request latencies initially rise. As new nodes are integrated, the gossip-based protocol quickly detects them and redistributes requests, leading to a clear reduction in latency.

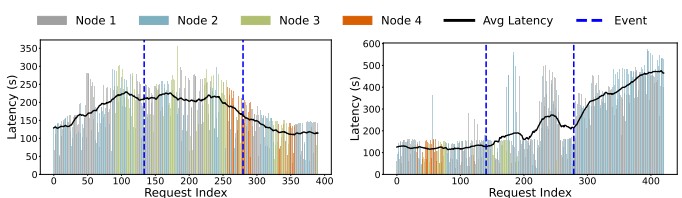

Figure 5: Request latency. Blue line indicates node join/leave events; black line shows the windowed average latency.

Conversely, the right panel in Figure 5 starts with four nodes and two leave the network sequentially. As the average load increases, the remaining nodes become increasingly saturated, resulting in a sharp rise in overall latency. These results demonstrate that WWW.Serve can dynamically adapt its workload distribution to both node arrivals and departures without a central coordinator, ensuring service continuity in unstable environments.

## 6.3 DUEL-AND-JUDGE EVALUATION

To evaluate the effectiveness of the duel-and-judge mechanism, we construct a small-scale network with four types of nodes: Qwen3 0.6B, Qwen3 4B, Qwen3 8B, and a random generator producing

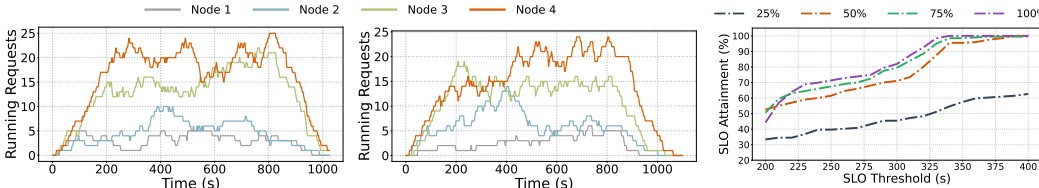

Figure 7: Left: Number of running requests under different stake amounts $(1, 2, 3, 4)$. Middle: Number of running requests under different acceptance frequencies $(0.25, 0.5, 0.75, 1.0)$. Right: SLO attainment under different offloading frequencies $(0.25, 0.5, 0.75, 1.0)$.

nonsensical responses. Each type has two replicas to mitigate randomness from single instances. We set the duel rate to $20\%$, with $k = 3$ judges per duel request.

Figure 6 (left) shows the evolution of credits: High-quality nodes (8B and 4B) steadily accumulate credits, while weaker nodes (0.6B) show only modest growth. Random generators are promptly penalized, experiencing continuous credit degradation. Figure 6 (right) highlights duel outcomes, where high-quality nodes secure substantially more victories. These results confirm that the duel-and-judge mechanism effectively distinguishes high-quality contributors from weak or malicious ones.

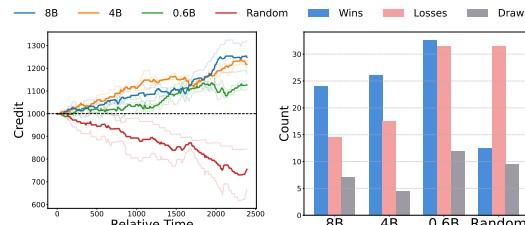

Figure 6: Evolution of credits (left) and duel outcomes (right) for different types of nodes.

We emphasize that the 20% duel rate used here is purely for experimental convenience, enabling rapid credit convergence and a clear observation of credit dynamics within a short time horizon (90 minutes in our experiment). A detailed analysis of the overhead introduced by the duel-and-judge mechanism is provided in Appendix F.

### 6.4 Ablation of Policies

We conduct an ablation to examine how user-level parameters (stake amount, request acceptance, and offloading frequency) affect workload allocation and global SLO attainment.

We first varied stake amounts and acceptance frequencies across nodes and monitored their local request queues. Requests were uniformly issued by a dedicated requester-only node. As shown in Figure 7 (left and middle), nodes with higher stake or higher acceptance frequency handle a larger share of delegated requests. This demonstrates that the PoS-based scheduling faithfully reflects user-level policies, allowing nodes to actively control their participation. Next, we evaluated the effect of offloading frequency under sustained high request pressure. As illustrated in Figure 7 (right), increasing offloading improves SLO attainment by redistributing workloads from overloaded nodes. However, the benefit saturates at moderate offloading rates: the improvement between rates of $0.5$, $0.75$, and $1.0$ is marginal. Excessive offloading can even hinder long-term credit accumulation as nodes spend more credits to delegate requests. Overall, these results confirm that WWW.Serve's flexible policy framework allows service providers to regulate their participation and optimize both efficiency and credit dynamics, indicating substantial room for fine-tuning policies to better balance immediate performance and long-term incentives.

## 7 Conclusion

This paper presents WWW.Serve, a fully decentralized framework for trustless and collaborative LLM serving. Operating as an open, competitive market for computational resources, it enables anonymous participants to autonomously route requests, balance workloads, and provide high-quality services. Our experiments demonstrate comparable scheduling efficiency along with strong adaptivity to dynamic resources and flexible serving policies, highlighting WWW.Serve's potential as a scalable and privacy-preserving foundation for next-generation LLM services.

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

## A  IMPLEMENTATION

In this section, we detail the implementation of the core modules contained in WWW.Serve.

*Communication Manager:* is implemented using `ZeroMQ`, providing low-latency, asynchronous message passing between nodes. We adopt the `ROUTER` pattern, where each node binds to a fixed port to listen for incoming messages while simultaneously sending requests to peers. This design enables efficient bidirectional communication without relying on a centralized broker.

*Request Manager:* leverages an asynchronous queue (`AsyncQueue`) for local request buffering and scheduling. Incoming requests are timestamped and inserted into the queue, while outgoing requests are dynamically dispatched to eligible executors based on the Proof-of-Stake–based selection mechanism and user-specific rules.

*Model Manager:* supports a variety of LLM serving backends via `AsyncOpenAI` clients. Service providers only need to supply a base URL and API key, without exposing internal model details. Each node periodically collects metrics from its backend servers, including the number of active and queued requests and memory utilization, to support efficient request dispatching and balanced workload distribution.

*Experiment Configuration:* is specified in a dedicated YAML file, capturing all necessary parameters for a node to initialize WWW.Serve modules. Each file includes: (i) Server Parameters: communication IP, port, user-level policy (e.g., stake, offload frequency, accept frequency), and backend selection (e.g., SGLang, vLLM); and (ii) Models: paths to local or remote LLMs, base URL for API access, and API keys. Each model entry also specifies generation parameters (e.g., maximum tokens, temperature, top-p) and dispatch parameters (e.g., target memory utilization). These YAML files are automatically parsed by each node at startup, ensuring reproducibility and allowing fine-grained control over node behavior.

## B  EXPERIMENTAL SETTINGS

To comprehensively evaluate the scheduling efficiency of WWW.Serve in heterogeneous, dynamic environments, we designed four distinct experimental settings, summarized in Table 3. Each setting varies in the deployed language models, GPU types, and serving backends, covering a broad spectrum of realistic node capabilities. Our evaluation primarily relies on recent open-source reasoning LLMs, including the Qwen3 series (Yang et al., 2025), DeepSeek-Qwen (DeepSeek-AI, 2025), and LLaMA 3.1 (Touvron et al., 2024), and prompts are drawn from the OpenR1-Math-220k dataset (Open-R1-Team, 2025). Time-varying request patterns are simulated via piecewise Poisson arrival rates for each node, capturing both high- and low-load periods that differ across nodes. Due to the limited scale of our experiments, we employ a shared ledger instead of a full Credit Block Chain, simplifying implementation while preserving the essential dynamics of credit transactions.

| Node | Model | GPU | Backend | Request Schedule | | | |
|------|-------|-----|---------|------------|------------|------------|------------|
| | | | | Interval 1 | $1/\lambda_1$ | Interval 2 | $1/\lambda_2$ |
| *Setting 1* | | | | | | | |
| Node 1 | Qwen3 8B | ADA6000 | SGLang | 0–300s | 5 | 300–750s | 20 |
| Node 2 | Qwen3 8B | ADA6000 | SGLang | 0–750s | 20 | | |
| Node 3 | Qwen3 8B | ADA6000 | SGLang | 0–750s | 20 | | |
| Node 4 | Qwen3 8B | ADA6000 | SGLang | 0–450s | 20 | 450–750s | 5 |
| *Setting 2* | | | | | | | |
| Node 1 | Qwen3 8B | ADA6000 | SGLang | 0–300s | 4 | 300–750s | 20 |
| Node 2 | Qwen3 8B | ADA6000 | SGLang | 0–750s | 20 | | |
| Node 3 | Qwen3 4B | RTX3090 | SGLang | 0–750s | 30 | | |
| Node 4 | Qwen3 4B | RTX3090 | SGLang | 0–450s | 30 | 450–750s | 6 |
| *Setting 3* | | | | | | | |
| Node 1 | Qwen3 32B | 4×A100 | SGLang | 0–300s | 2 | 300–750s | 6 |
| Node 2 | Qwen3 8B | L40S | SGLang | 0–750s | 15 | | |
| Node 3 | DeepSeek-Qwen 7B | RTX3090 | vLLM | 0–750s | 30 | | |
| Node 4 | Llama3.1 8B | ADA6000 | vLLM | 0–450s | 15 | 450–750s | 5 |
| *Setting 4* | | | | | | | |
| Node 1 | Llama3.1 8B | L40S | vLLM | 0–750s | 9 | | |
| Node 2 | Llama3.1 8B | L40S | vLLM | 0–450s | 6 | 450–750s | 12 |
| Node 3 | DeepSeek-Qwen 7B | ADA6000 | vLLM | 0–300s | 6 | 300–750s | 12 |
| Node 4 | DeepSeek-Qwen 7B | ADA6000 | vLLM | 0–450s | 12 | 450–750s | 6 |
| Node 5 | Qwen3 4B | RTX4090 | SGLang | 0–750s | 12 | | |
| Node 6 | Qwen3 4B | RTX4090 | SGLang | 0–450s | 10 | 450-750s | 20 |
| Node 7 | Qwen3 4B | RTX3090 | SGLang | 0–300s | 20 | 300–750s | 10 |
| Node 8 | Qwen3 4B | RTX3090 | SGLang | 0–300s | 20 | 300–750s | 10 |

Table 3: Experimental configurations correspond to Figure 4 (left to right) and Table 2. Each setting specifies the deployed model, GPU type, serving backend, and the time-varying request schedule for all nodes. The Interval columns specify the time ranges, and the corresponding $1/\lambda$ columns denote the expected inter-arrival time (in seconds) used for Poisson request generation, i.e., request inter-arrival times distributed as $Poi(\lambda)$.

All nodes are configured with consistent policy parameters, including offload frequency (80%), acceptance frequency (80%), target utilization (70%), and generation parameters such as maximum token length (8192), temperature (0), and top-p sampling (0.95). These standardized settings ensure comparability and reproducibility across heterogeneous nodes while enabling a systematic evaluation of the effects of resource diversity and dynamic workloads on scheduling efficiency, latency, and SLO attainment.

## C  USE OF LARGE LANGUAGE MODELS

We used large language models (LLMs) solely as language editing tools to polish grammar, improve readability, and refine the academic style. All research ideas, methods, experiments, and analyses were independently conceived and conducted by the authors without assistance from any LLMs.

## D  TERMINOLOGY CLARIFICATION

Table 4 provides definitions of several key concepts referenced in this paper.

| Concept | Meaning in Our System |
|---|---|
| **Node** | A service provider participating in the network. Each node hosts its own LLM server and can process inference requests. |
| **User Request** | A request submitted by the users of a given node. The node may either execute it locally or offload part of the load to our system when resources are constrained. |
| **Delegated / Offloaded Request** | A request forwarded from another node. Upon receiving such a request, a node may choose to execute it or further offload it based on its own policy. |
| **User-Level Policy** | Node-specific policies governing how the node interacts with its own users. Examples include: when to offload, whether to accept delegated requests, prioritization of local users, and whether users permit offloading. These policies are fully controlled by each node. |
| **System-Level Policy** | Global coordination rules of our system, including PoS-based scheduling, gossip-driven protocol, and the duel-and-judge mechanism. These govern decentralized cooperation among anonymous nodes. |

Table 4: Clarification of terminology.

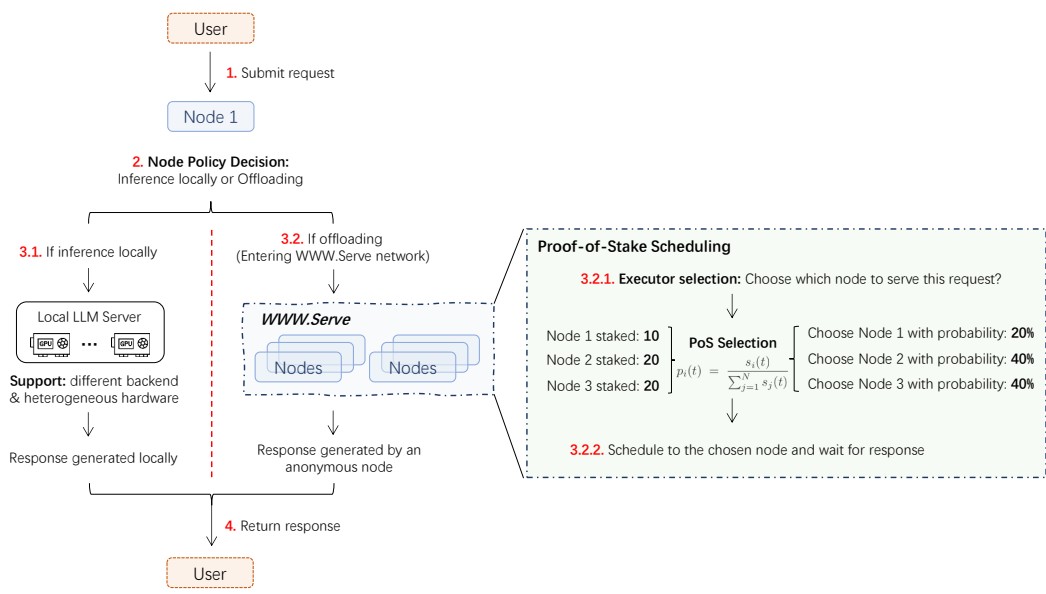

Figure 8: End-to-end workflow of a single user request, including local execution or remote offloading via PoS-based scheduling.

# E    SUPPLEMENTARY SYSTEM DETAILS

In this section, we provide additional illustrations that complement the descriptions in Section 3 and Section 4, focusing on two key components of WWW.Serve: (i) the end-to-end workflow of processing a single user request, and (ii) the gossip-driven protocol for peer synchronization.

## E.1    REQUEST PROCESSING WORKFLOW

Figure 8 presents the end-to-end workflow of a node handling a user request. Upon receiving a query (Step 1), the node determines whether to execute it locally or offload it to the network (Step 2).

**Local execution (Step 3.1):** Nodes may host local language models using diverse runtimes (e.g., vLLM, SGLang) on heterogeneous devices. WWW.Serve abstracts these differences through a uni-

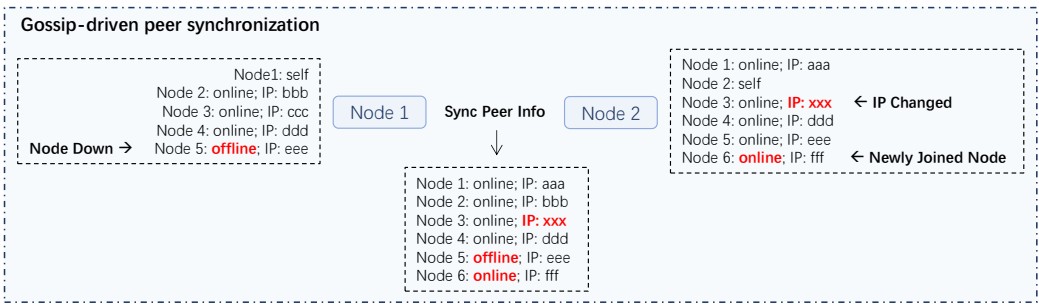

Figure 9: Gossip-driven peer synchronization. During each gossip round, nodes exchange local peer views, allowing updated information to propagate diffusively throughout the network.

fied inference interface, allowing heterogeneous hardware and software stacks to participate without modifications to global collaboration mechanisms.

**Remote execution (Step 3.2):** If offloading is selected, the node samples a trustworthy executor through our PoS-based scheduler (Step 3.2.1), where each peer's sampling probability is proportional to its staked credit. Once an executor accepts the task, the request is forwarded for processing and the generated response is returned to the origin node.

### E.2 GOSSIP-DRIVEN PEER SYNCHRONIZATION

Figure 9 shows an example gossip synchronization between two nodes. Each node maintains a local view of peer availability, including identifiers, online/offline status, and communication endpoints. During a gossip round, two nodes exchange their current views and reconcile any discrepancies, for instance, peers that have gone offline (Node 5), updated their network addresses (Node 3), or newly joined (Node 6). Repeated lightweight pairwise exchanges allow updates to diffuse across the network and converge quickly, without requiring any central coordinator.

# F  OVERHEAD OF DUEL-AND-JUDGE MECHANISM

This section presents a theoretical analysis of the overhead introduced by the duel-and-judge mechanism, followed by an empirical evaluation of latency and SLO attainment under different duel rates.

We first quantify the incremental request load. Let:

- $N$: total number of user requests across all nodes;
- $\alpha$: request delegation rate ($\alpha N$ requests are offloaded for remote inference);
- $p_d$: duel rate (a fraction $p$ of delegated requests are selected as duel requests);
- $k$: number of judges per duel.

Each duel request triggers one challenger inference and $k$ judge evaluations, contributing $(1 + k)$ additional requests. Thus, the expected number of extra requests introduced by the duel-and-judge mechanism is

$$N\alpha\,p\,(1 + k),$$

which remains modest compared to the overall serving workload.

To empirically evaluate the effect of duel rate on system performance, we conduct an ablation study using four nodes, with $k = 2$ judges per duel. Requests are uniformly issued by a dedicated requester-only node. This configuration intentionally imposes higher load than typical deployments: fewer nodes yet multiple judges per duel amplify the relative overhead. As shown in Figure 10, duel probabilities of 5%, 10%, and 25% yield nearly identical latency CDFs and SLO attainment curves, indicating that moderate duel rates introduce minimal overhead.

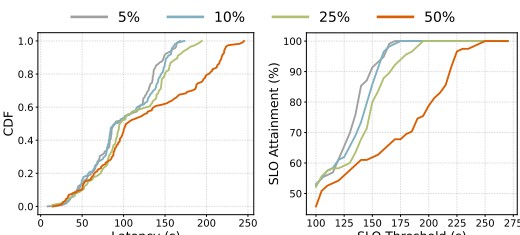

Figure 10: Latency CDF (left) and SLO attainment (right) for different duel rates.

# G  PERFORMANCE OF PRODUCTION BLOCKCHAIN SYSTEMS

In WWW.Serve, the blockchain-based credit ledger can be instantiated with any suitably provisioned blockchain, serving primarily to maintain a tamper-resistant record of credit transactions in a fully decentralized network. Consequently, to contextualize its scalability and efficiency, we summarize the performance of several mature blockchain systems, providing representative throughput and latency metrics that WWW.Serve would inherit when built on similar foundations.

| System | Throughput (TPS) | Latency (s) |
|---|---|---|
| **Hyperledger Fabric** Androulaki et al. (2018) | $\sim$ 3,500 | $< 1$ |
| **FastFabric** Gorenflo et al. (2019) | $\sim$ 20,000 | $< 1$ |
| **Aptos** Aptos (2022) | $\sim$ 20,000 | $\sim$ 1.25 |
| **Zaptos** Xiang et al. (2025) | $\sim$ 20,000 | $\sim$ 0.75 |

Table 5: Performance of representative blockchain systems. TPS: transactions per second; Latency: the time between when a transaction is sent and when it's added to the blockchain.

