# OpenReview forum: "WWW.Serve: A Decentralized Framework for Collaborative LLM Serving"
_ICLR.cc/2026/Conference — Submitted to ICLR 2026_

### Official Review · Reviewer_DmSJ · 2025-10-23

**Soundness:** 2
**Presentation:** 1
**Contribution:** 2
**Rating:** 2
**Confidence:** 2

**Summary:**

This paper presents WWW.Serve, a decentralized framework that interconnects LLM servers worldwide. It integrates three core design components that preserve service-provider anonymity and privacy while enabling self-organizing request dispatch, dynamic load balancing, and autonomous control over resources and policies. The work shows good originality and includes extensive experiments. However, the presentation quality is only fair, leaving several points that require revision or clarification; parts of the theoretical analysis lack rigor, and the novelty appears limited as currently presented.

**Strengths:**

1. There are originality merits in this paper.

2. The experimental evaluation is sufficient and comprehensive,.

**Weaknesses:**

1. While the topic is timely, the manuscript’s contribution is not clearly demonstrated. It largely reads as a combination of distributed learning, blockchain, and game theory without convincingly motivating gaps in prior work. The Introduction and Background sections do not explain why existing approaches are insufficient or how this work advances the state of the art.

2. The paper assumes a linear “win-probability–quality” mapping without derivation or parameterization from standard pairwise-comparison models; it overclaims convergence under replicator dynamics without Lyapunov or invariant-set analysis; key assumptions are not enumerated, results are not presented in theoremized form (i.e., Assumptions, Lemma/Proposition, Theorem and Proof), and parameter identifiability/sensitivity/robustness analyses are missing.

3. The presentation needs substantial editing; several passages are confusing or contain errors. Specific issues include:

a) Define Service Level Objective (SLO) on first use in the abstract.

b) Figure 1 and its caption are inconsistent: the caption states that the upper/lower panels show the workflow and the architecture, but the figure uses arrows to imply relationships among mixed elements. Consider splitting into separate figures.

c) The Figure 1 caption in §3.2 is unclear.

d) Several terms are undefined or used only once without explanation (e.g., “delegated requests,” “predefined threshold,” “gossip-driven”).

e) The contribution statement (line 83) lists three core mechanisms—credit-based transaction system, gossip-driven protocol, and duel-and-judge mechanism—but §4 clearly introduces only the first and third; the “gossip-driven protocol” is left undefined or uses terminology inconsistent with earlier text.

**Questions:**

1. Why is it named “WWW.Serve”? What does it have to do with what is presented and studied in this paper?

2. What does “delegated requests” mean (line 265)?

3. What is the relation between Proof-of-Stake (PoS) mechanism and the Dual-and-Judge mechanism?

4. What does gossip-driven mean in this context?

5. The discussion of “Executor selection and trust establishment” needs further clarification.

6 . The discussion of “Execution across heterogeneous backends” also needs further clarification.

---

> ### Author Response · Authors · 2025-11-25
> **Response to Reviewer DmSJ (part 1)**
>
> Thank you very much for your thoughtful review and constructive suggestions. We are glad that the reviewer recognized the **originality of our work** and found the **experimental evaluation sufficient and comprehensive**. We have tried to address your questions carefully. We hope the reviewer will consider raising your score in light of our response.
>
> ## W1: Unclear contribution and insufficient motivation relative to prior work
>
> We have **revised Abstract, Introduction, and Related Work** ($\color{blue}{\text{Section 1-2, Pg. 1-3}}$) to clarify our motivation, highlight gaps in prior work, and show how our approach advances the field. The core ideas are summarized below.
>
> Centralized LLM serving achieves strong efficiency but limits scalability, and leaves substantial scattered GPU resources underutilized. Decentralized serving could potentially address these limitations, but existing frameworks **predominantly emphasize the rights and protections of users and the cooperative aspect among GPU providers** while **overlooking the inherent competitive dynamics**, imposing substantial constraints on GPU providers, such as requiring them to accept excessive platform-level oversight and to execute all assigned requests with fixed software stacks on fixed hardware configurations. These assumptions **conflict with real-world conditions**, where GPU providers are **self-interested, experience fluctuating workloads, and need freedom to decide how they participate**.
>
> To bridge this gap, we propose WWW.Serve, a decentralized framework that acts like an **open, competitive market**, allowing GPU holders to decide when, under what policies, and with what resources they join the decentralized network. Our key contributions are:
> 1. **Trustworthy market-driven trade of computational capacity**: a credit-based transaction system that facilitates the workload balancing among anonymous participants.
> 2. **Incentive-aligned service quality**: a well-designed reward mechanism that incentivizes providers to deliver higher-quality services, including faster hardware, more user-oriented scheduling policies, better serving systems, and higher-quality models.
> 3. **Flexible participation**: resource providers freely choose when, how, and with what resources to join, while preserving anonymity.
>
> Empirically, we show that WWW.Serve incentivizes higher-quality services to obtain greater profit, while improving global SLO (service-level-objective) attainment by up to 1.5x and lowers latency by 27.6%. These results highlight WWW.Serve as a promising foundation for real-world, decentralized LLM serving.
>
> ## W2, W3: Presentation and Clarity of Theoretical and Experimental Details
>
> We have **revised Abstract, Introduction, and Related Work** ($\color{blue}{\text{Section 1-2, Pg. 1-3}}$) to clarify our motivation, highlight gaps in prior work, and show how our approach advances the field. We have also **formalized the game-theoretic analysis** in $\color{blue}{\text{Section 5, Pg. 7-8}}$ with theorem statements to provide a clearer and more rigorous presentation, provided precise **definitions of several key concepts** referenced in our work in $\color{blue}{\text{Appendix D, Pg. 15-16}}$, and updated **new figures to illustrate our framework** in $\color{blue}{\text{Appendix E, Pg. 16-17}}$. These revisions directly address the reviewer’s concerns in Q2, Q3, and Q4.
>
> ## Q1： Why is our framework named “WWW.Serve”?
>
> The name “WWW.Serve” reflects our design goal: a decentralized framework for interconnecting LLM servers across the globe in a trustless, collaborative manner. Our system aims to support **Internet-scale multi-LLM serving**, where independently operated nodes cooperate through decentralized scheduling. The name emphasizes this vision of a **worldwide, service-level network** rather than a single cluster or provider, directly matching the architecture and behaviors studied in the paper.
>
> ## Q2: What does “delegated requests” mean (line 265)?
>
> “Delegated request” represents a request **forwarded from another node**. Upon receiving such a request, a node may choose to execute it or further offload it based on its own policy. We further provide precise **definitions of several key concepts** referenced in our work in $\color{blue}{\text{Appendix D, Pg. 15-16}}$.

---

> ### Author Response · Authors · 2025-11-25
> **Response to Reviewer DmSJ (part 2)**
>
> ## Q3: What is the relation between Proof-of-Stake (PoS) mechanism and the Dual-and-Judge mechanism?
>
> The Proof-of-Stake (PoS) mechanism and the duel-and-judge mechanism play complementary roles in establishing trust in our fully decentralized network.
>
> 1. **PoS scheduling determines who is selected as an executor**. When a node decides to offload a request, candidate executors are sampled with probability proportional to their staked credit. This ensures that nodes with a history of good behavior are more likely to receive requests, while still allowing low-stake nodes to participate.
> 2. **Duel-and-Judge mechanism determines how credit evolves over time**. To make credit a meaningful proxy for trust, we periodically evaluate nodes through duel requests: the same request is assigned to two candidate nodes, and independent judges assess their responses. High-quality responses gain credit, while poor-quality responses lose credit. This redistributes credit according to demonstrated performance rather than self-reported claims or static reputation.
>
> Together, the two mechanisms form a closed loop: **Duel-and-Judge mechanism continuously reshapes credit based on observed quality, and PoS scheduling uses that credit distribution to guide executor selection**. This coupling ensures that credit reflects actual service quality and that trustworthy nodes naturally become more influential over time, without requiring any centralized authority.
>
> ## Q4: What does gossip-driven mean in this context?
>
> We have added a figure in $\color{blue}{\text{Appendix E, Pg. 16-17}}$ to **detail our gossip-driven protocol**. In our context, gossip-driven synchronization means that **each node periodically exchanges its known peer-state information with a randomly selected subset of neighbors**, rather than relying on any central coordinator. As illustrated in the updated figure, a node maintains a local view of peer availability (e.g., node ID, online/offline status, communication IP address). During a gossip round, Node 1 and Node 2 exchange their local views; discrepancies, such as Node 5 going offline, Node 3 changing IP address, or Node 6 newly joining, are reconciled on both sides. Repeating this lightweight pairwise exchange across the network ensures that updated information spreads diffusively and converges rapidly, even at large scale. This design provides robustness to churn and avoids the communication bottlenecks of global broadcasts or centralized monitoring.
>
> ## Q5, Q6: The discussion of “Executor selection and trust establishment” and “Execution across heterogeneous backends” needs further clarification.
>
> We have added a detailed figure in $\color{blue}{\text{Appendix E, Pg. 16-17}}$ **illustrating the end-to-end workflow of a single user request**, which clarifies both the executor selection and trust establishment process as well as execution across heterogeneous backends.
>
> After a user submits an LLM query to a node (Step 1), the node determines whether to execute it locally or delegate it to the WWW.Serve network (Step 2):
>
> 1. **If offloading**, the node selects a trustworthy executor among the anonymous peers via a **PoS-based scheduler** (Step 3.2.1), where each peer’s probability of being sampled is proportional to its staked credit. This prioritizes nodes with a history of high-quality performance while still giving low-stake nodes a chance to be selected. Once an executor is chosen and accepts the task, the request is forwarded for remote execution and the response is returned to the originator.
> 2. **If locally inferencing**, our updated figure highlights the execution path in Step 3.1. Our node design decouples scheduling logic from the underlying serving engines. In practice, nodes may host models using **different runtimes** (e.g., vLLM, SGLang, MLC-LLM) on **heterogeneous computational resources**. Our node normalizes these differences by exposing a consistent interface for inference execution. This design ensures that heterogeneous hardware and software stacks can be seamlessly incorporated into WWW.Serve without requiring any modifications to the global scheduling, staking, or quality-assessment mechanisms.

---

### Official Review · Reviewer_SRSf · 2025-10-31

**Soundness:** 3
**Presentation:** 3
**Contribution:** 3
**Rating:** 6
**Confidence:** 2

**Summary:**

This paper proposes WWW.Serve a fully decentralized, multi-model LLM serving framework. The design combines (i) a blockchain-inspired credit ledger with staking to incentivize reliable service and enable tamper-resistant accounting, (ii) Proof-of-Stake–based selection for routing/assignment, and (iii) a duel-and-judge mechanism in which a small fraction of delegated requests are executed twice and peer-judged to continuously calibrate node reputation and quality. The implementation uses asynchronous messaging and supports heterogeneous backends and models. Empirically, across four heterogeneous settings, the decentralized scheduler consistently outperforms single-node service and closely matches (sometimes slightly surpasses) a centralized scheduler in SLO attainment, while also adapting to dynamic node joins/leaves.

**Strengths:**

1. It is an interesting attempt for decentralized LLM, for example, the credit ledger + staking design provides an auditable, tamper-resistant accounting path without a trusted coordinator; and the duel-and-judge procedure encourages high-quality service and penalizes poor or malicious behavior, avoiding privileged verification committees.

2. The paper provides an in-depth system design for decentralized LLM infra design, such as Asynchronous messaging (ZeroMQ ROUTER), backend-agnostic integration (e.g., SGLang/vLLM), and reproducible YAML configs.

**Weaknesses:**

1.  My major question is that the evaluation is limited in scale and uses a shared ledger instead of a full Credit Block Chain, leaving open questions about ledger consensus/throughput and communication overheads at 10²–10³ nodes. In real-world applications, such large scale communication overheads may be unavoidable. Strengthening Section B with either simulation or larger-scale measurements of ledger sync and gossip convergence would increase confidence.

2.  To better position the work, adding at least one controlled, end-to-end comparison in an overlapping regime (shared model/backbone and request mix)is necessary, such as evaluation compared against Petals[1] (volunteer P2P collaborative inference for fixed LLMs), DeServe[2] (decentralized offline serving; reports 6.7×–12.6× throughput gains under high-latency networks), and/or GenTorrent[3] (overlay-based serving; reports >50% latency reduction vs. a non-overlay baseline).

References (for Weakness 2)

[1] Borzunov, Alexander, et al. "Petals: Collaborative inference and fine-tuning of large models." ACL 2023 (demo).

[2] Wu, Linyu, et al. "DeServe: Towards Affordable Offline LLM Inference via Decentralization." arXiv preprint arXiv:2501.14784 (2025).

[3] Fang, Fei, et al. "GenTorrent: Scaling Large Language Model Serving with An Overley Network." arXiv preprint arXiv:2504.20101 (2025).

**Questions:**

1. Under churn (joins/leaves), what are the per-transaction compute/bandwidth costs and the convergence times for both ledger synchronization and gossip when moving from the shared ledger used here to the full Credit Block Chain? Any preliminary numbers or simulations?

2. How do duel probability and #judges impact SLO attainment and tail latency (p95/p99)? A brief ablation quantifying overhead vs. quality-assurance benefit would be helpful.

---

> ### Author Response · Authors · 2025-11-25
> **Response to Reviewer SRSf**
>
> Thank you very much for your thoughtful review and constructive suggestions. We are glad that the reviewer found our **system design in-depth and interesting**. We have tried to address your questions carefully. We hope the reviewer will consider raising your score in light of our response.
>
> ## W1, Q1: The evaluation is limited in scale and uses a shared ledger instead of a full Credit Block Chain, leaving open questions about ledger consensus/throughput and communication overheads at 10²–10³ nodes.
>
> We would like to clarify that the main focus of our work is not on blockchain performance, but on designing a decentralized multi-LLM serving system. Our ledger implementation is **orthogonal to our contribution** and can be instantiated with **any suitably provisioned blockchain technology**, only for providing a tamper-resistant record of credit transactions under the fully decentralized network. Consequently, the throughput and latency of the ledger can be directly supported by existing blockchain systems. We summarize the **performance of representative production blockchains** in $\color{blue}{\text{Appendix G, Pg. 18}}$:
>
> | System                  | Throughput (TPS) | Latency (s) |
> |-------------------------|---------------------------|----------------|
> | **Hyperledger Fabric** | ~3,500            | <1              |
> | **FastFabric**               | ~20,000          | <1              |
> | **Aptos**                       | ~20,000          | ~1.25         |
> | **Zaptos**                     | ~20,000          | ~0.75         |
>
> Regarding communication overhead in our gossip protocol, each node exchanges only a bounded metadata payload (node ID, status, IP, credit info) with **O(1) neighbors per cycle**. This leads to O(kN) total message exchanges per round for a constant fan-out k. Such protocols are well-understood to scale efficiently to thousands of nodes and are already used in large production systems with significantly heavier metadata:
> - **Cassandra**’s gossip protocol maintains cluster membership and failure detection across 1000+ nodes.
> - **Ethereum’s P2P network and Consul** use a similar partial-view mesh for state propagation.
>
> ## W2: Adding comparison: Petals, DeServe, and GenTorrent.
>
> We conducted a direct **comparison between Petals and WWW.Serve** on inference efficiency. Under identical hardware and model settings, WWW.Serve achieves substantially lower latency and higher throughput for generation lengths of 128, 1024, and 4096 tokens, as shown in the table below:
>
> | System | Gen length | Latency (s) | Throughput (token/s) |
> |------------|----------------|-----------------|-----------------------------|
> | Petals   | 128             |  17.8931     | 7.15                          |
> | WWW.Serve   | 128   |  **2.8366**       | **45.12**                         |
> | Petals   | 1024           |  142.0986   |  7.21                          |
> | WWW.Serve  | 1024  |  **21.1692**     |   **48.37**                       |
> | Petals   | 4096           |  542.2454    | 7.55                          |
> | WWW.Serve  | 4096  |  **76.4542**     |  **48.60**                        |
>
> Additionally, we would like to emphasize that Petals and WWW.Serve target fundamentally different problem settings. Petals focuses on **collaborative inference for a single fixed LLM** across many volunteer devices, enabling users with limited local resources to **run an otherwise unhostable LLM**. In contrast, WWW.Serve is a **fully decentralized multi-LLM serving framework** that focuses on **cross-LLM scheduling** across an open network of heterogeneous nodes. From this perspective, Petals corresponds to the intra-node serving layer in our architecture and could, in principle, be used within a single WWW.Serve node. Our system operates at a different architectural layer, coordinating multiple model servers rather than replacing the collaborative inference mechanism within a node.
>
> ## Q2: How do duel probability and #judges impact SLO attainment and tail latency (p95/p99)?
>
> We appreciate the reviewer’s question regarding the impact of duel probability and number of judges on SLO attainment and tail latency. Conceptually, each duel introduces one challenger inference plus k judge requests, so a **duel probability of p** increases the total offloading volume by roughly **p(1+k)**, without affecting locally served requests. The resulting overhead is small relative to overall LLM serving workload. A **detailed ablation study** in $\color{blue}{\text{Appendix F, Pg. 18}}$ presents both theoretical analysis and empirical evaluation of **latency CDF and SLO attainment** for different duel rates, confirming that moderate duel rates (5%-20%) introduce minimal overhead.

---

### Official Review · Reviewer_spMp · 2025-11-06

**Soundness:** 3
**Presentation:** 3
**Contribution:** 2
**Rating:** 4
**Confidence:** 3

**Summary:**

This paper introduces WWW.Serve, a decentralized framework for collaborative LLM serving. The goal is to address the limitations of centralized services, such as restricted scalability and privacy risks. The system interconnects distributed and anonymous LLM servers in a peer-to-peer network, enabling them to share computational resources and balance workloads. The core of WWW.Serve consists of three key mechanisms: (1) a blockchain-inspired credit system for trustless transactions, (2) a gossip-driven protocol for dynamic peer discovery and synchronization, and (3) a "duel-and-judge" mechanism to ensure service quality. The paper provides a game-theoretic analysis suggesting the system converges to a high-quality equilibrium and presents empirical results demonstrating WWW.Serve's performance.

**Strengths:**

- **Significant and Timely Problem**: The paper tackles a highly relevant real-world challenge in the age of LLMs. Building a decentralized, scalable, and privacy-preserving serving infrastructure could have a transformative impact on the field.
- **Comprehensive System Design**: The proposed WWW.Serve framework is well-thought-out, with its three core mechanisms working in concert to address the key challenges of trust, coordination, and quality control in a decentralized setting.
- **Theoretically-Grounded Analysis**: The inclusion of a game-theoretic analysis (Section 5) provides a theoretical foundation for the system's incentive structure, arguing for its convergence towards a high-quality service equilibrium.

**Weaknesses:**

**1. Idealized Evaluation vs. Real-World Complexity:** The experiments, while internally consistent, are conducted in a "laboratory" setting that abstracts away critical real-world challenges.

  - ***Scale***: The system is evaluated on a very small number of nodes (4-8). This is insufficient to validate the scalability of a P2P system named "WWW.Serve" or to reveal potential issues with the gossip protocol at scale.
  - ***Network Conditions***: The experiments appear to assume a high-bandwidth, low-latency network. Real-world systems must contend with geographic distribution, variable bandwidth, and network partitions.
  - ***Simplified Economic and Operational Dynamics***: The game-theoretic model (Section 5) presumes nodes are rational actors with a static quality q_i, overlooking more sophisticated strategies and dynamics. Additionally, the paper has not dicussed the "cold-start" problem: How are initial credits distributed, and what mechanism bootstraps the network by attracting the first cohort of high-quality providers?

**2. Unacknowledged Overheads and Practicality:** The paper downplays the significant overhead of its core mechanisms.

  - The "duel-and-judge" mechanism is described as applying to a "small fraction" of requests (line 265), but the experiments use a 20% duel rate (line 448). This implies a substantial, non-trivial overhead, which is neither quantified nor justified.
  - Similarly, the performance impact of the "blockchain-inspired" ledger is glossed over. High-frequency transactions in a large-scale network could create a severe bottleneck, but the paper simplifies this by using a "shared ledger" in experiments (line 695-696).

**3. Potential for Re-Centralization:** The paper's own game-theoretic analysis hints at a potential paradox. Equation (1) (ṗ_i ∝ p_i(∆_i - ∆)) describes a "rich-get-richer" dynamic. While the authors frame this as "quality wins," it also implies that nodes with initial advantages (e.g., a large corporation with superior hardware and lower costs) will see their stake share grow exponentially, potentially leading to market concentration and defeating the very purpose of decentralization. This critical long-term dynamic and its negative implications are not discussed.

**4. Unresolved Privacy and Security Issues:** The paper claims to enhance privacy, but it primarily focuses on provider anonymity. It critically fails to address the glaring issue of user data privacy. Sending user prompts (which can be highly sensitive) in plaintext to anonymous, untrusted nodes is a major security risk. The framework lacks any mechanism to protect user data.

**Questions:**

Following up on the points raised in the Weaknesses:

1. Could the authors provide a more detailed analysis of the system's overhead? Specifically, what is the justification for the 20% duel rate, and what is its expected impact on overall system throughput and cost?
2. Regarding the game-theoretic analysis in Section 5, while it predicts convergence to high quality, doesn't Equation (1) also predict a strong market concentration dynamic ("rich-get-richer")?
3. The paper highlights "privacy" as a key benefit. What mechanisms are envisioned to protect sensitive user prompts when they are processed by anonymous, untrusted nodes in the network?
4. The experimental validation is conducted in an idealized setting. How would the system's performance, particularly the request routing and gossip protocol, be affected by real-world network conditions such as high inter-node latency and limited bandwidth?

---

> ### Author Response · Authors · 2025-11-25
> **Response to Reviewer spMp (part 1)**
>
> Thank you very much for your thoughtful review and constructive suggestions. We are glad that the reviewer found our **system design comprehensive and theoretically analysis grounded**. We have tried to address your questions carefully. We hope the reviewer will consider raising your score in light of our response.
>
> ## W1, Q4:  Idealized Evaluation vs. Real-World Complexity
>
> We agree that evaluating system performance under real-world conditions is important. In the following, we analyze four aspects: the effect of **network latency**, the **efficiency of the ledger and gossip protocol**, the **“cold-start” problem**, and the **network bootstrap mechanism**.
>
> 1. To assess the effect of inter-node latency, we **injected synthetic delays** sampled from an exponential distribution into the request-sending path and measured end-to-end performance. The results, summarized below, show that the system remains stable when the injected latency is within 300 ms, while performance gradually degrades as latency increases beyond this range, as expected.
>
>     | Synthetic Delay (s) | Avg Latency (s) | Avg Throughput (token/s) | p95 Latency (s) | p99 Latency (s)  |
>     |------------------------|-----------------------|-----------------------------------|----------------------|-----------------------|
>     | 0                          | 75.1897             | 87.86                                | 120.9926           | 129.2882            |
>     | 50                        | 73.9624             | 87.86                                | 114.5052           | 117.7532            |
>     | 100                      | 74.5123             | 86.96                                | 116.4965           | 124.5898            |
>     | 300                      | 76.2464             | 86.25                                | 118.4419           | 132.2803            |
>     | 500                      | 77.9760             | 84.50                                | 117.2979           | 129.9174            |
>     | 1000                    | 79.1607             | 80.14                                | 117.3263           | 122.5198            |
>
> 2. Additionally, we would like to clarify that **optimizing the efficiency of request scheduling and gossip protocol is not the main focus of our work**. Both components leverage well-established, production-tested mechanisms whose scalability is already known:
>
>     - Our ledger implementation is **orthogonal to our contribution** and can be instantiated with **any suitably provisioned blockchain technology**, only for providing a tamper-resistant record of credit transactions under the fully decentralized network. Therefore, the throughput and latency of the ledger directly inherit the performance of the chosen backend. We summarize the **performance of representative production blockchains** in $\color{blue}{\text{Appendix G, Pg. 18}}$:
>
>         | System                  | Throughput (TPS) | Latency (s) |
>         |-------------------------|---------------------------|----------------|
>         | **Hyperledger Fabric** | ~3,500            | <1              |
>         | **FastFabric**               | ~20,000          | <1              |
>         | **Aptos**                       | ~20,000          | ~1.25         |
>         | **Zaptos**                     | ~20,000          | ~0.75         |
>
>     - Our gossip protocol exchanges only a bounded metadata payload (node ID, status, IP, credit info) with **O(1)** neighbors per cycle, yielding **O(kN)** total message exchanges per round for a constant fan-out k. This design follows well-understood gossip protocols that scale efficiently to thousands of nodes and are already used in large deployments handling far heavier metadata:
>         - **Cassandra**’s gossip protocol maintains cluster membership and failure detection across 1000+ nodes
>         - **Ethereum’s P2P network and Consul** use a similar partial-view mesh for state propagation.
>
> 3. The cold-start problem can be mitigated using mechanisms analogous to those in existing real-world blockchain systems. For example, in **Cardano or Algorand**, initial stake is nominally assigned to seed nodes, and the network gradually forms a stable distribution of stake based on participation and performance.
>
> 4. Finally, regarding the bootstrap stage, we note that **participating in our system is inherently beneficial**. Nodes with idle resources can accumulate credit by serving delegated requests, effectively turning spare capacity into future utility. When under heavy load, nodes can offload requests to the network to reduce congestion and improve throughput. This flexible and low-risk participation model naturally attracts providers during bootstrap, as it enhances performance without requiring any upfront commitment.

---

> ### Author Response · Authors · 2025-11-25
> **Response to Reviewer spMp (part 2)**
>
> ## W2, Q1: Unacknowledged Overheads and Practicality
>
> We would like to clarify that the “20% duel rate” used in Subsection 6.3 is intentionally chosen for **fast credit convergence** in a controlled experimental setting. Our goal in this experiment is to clearly demonstrate the effectiveness of our duel-and-judge mechanism: high-quality nodes steadily accumulate credit while low-quality nodes lose credit. A relatively high duel rate accelerates this redistribution process, **making the credit dynamics observable** within a short number of rounds (90min in our experiment).
>
> In real deployments, such a high duel rate is unnecessary. Once the network stabilizes and nodes’ credit ranks reflect their long-term quality, duels become infrequent and serve only as periodic quality checks. The duel rate can also be made adaptive, e.g., using lower rates for nodes with long-standing reputation and higher rates only for newly joined nodes during their bootstrap period. Because each duel consists of **one additional inference request plus k judge inferences**, the resulting overhead remains small relative to the overall LLM serving workload.
>
> We have revised $\color{blue}{\text{Subsection 6.3, Pg. 10}}$ to **clarify the duel rate setting**, and updated **a detailed ablation study** in $\color{blue}{\text{Appendix F, Pg. 18}}$ providing both theoretical analysis and empirical evaluation of latency CDF and SLO attainment for different duel rates, which confirms that moderate duel rates (5%–20%) introduce minimal overhead.
>
> ## W3, Q2: Potential for Re-Centralization (“rich-get-richer”)
>
> We thank the reviewer for highlighting the potential re-centralization issue. Our analysis in Section 5 mainly focuses on the **payoff generated from serving delegated requests**, showing that high-quality nodes receive positive net payoff while lower-quality nodes face penalties. This mechanism ensures that the network does not become dominated by low-quality providers: nodes are incentivized to maintain higher service quality in order to achieve more net gains. We have also **formalized the game-theoretic analysis** in $\color{blue}{\text{Section 5, Pg. 7-8}}$ with theorem statements to provide a clearer and more rigorous presentation.
>
> It is important to note, however, that from the full system view, **high quality does not guarantee future credit growth**. Each node also **spends credits when offloading its own queries** to others in order to boost throughput under high load. Thus, the net credit trajectory of a node is determined by the balance between (i) revenue from serving delegated requests and (ii) expenditure from outsourcing its own tasks, which depends on both quality and economic strategy (e.g., how aggressively a node chooses to offload).
>
> We acknowledge that a rich-get-richer effect may arise under an **extreme but conceptually simple scenario**: a node consistently produces high-quality responses while rarely offloading its own queries. Such a node effectively becomes a net provider of service to the rest of the network. In that case, it continuously contributes useful work to the system and accumulates credits without incurring comparable expenditures. While this behavior reflects genuine value creation rather than a structural bias in the mechanism, it nonetheless reintroduces centralization. We recognize this as an important design consideration. Developing complementary mechanisms is a promising direction for future work.
>
> ## W4, Q3: Besides service provider anonymity, how to protect user data privacy?
>
> We would like to clarify that our framework does not require service providers to expose user requests to the decentralized network. Each node’s **own local policy determines whether a request is delegated**, and this decision is **fully transparent** and under the complete control of the service provider. If a provider chooses not to offload, the request is processed entirely on the local server and never leaves the node. Our system only performs scheduling and inference on requests that a provider explicitly decides to delegate.
>
> As a result, user data privacy is primarily governed by the **agreement between the user and the service provider** rather than by our decentralized infrastructure. The framework does not impose any mandatory data-sharing behavior, nor does it override the provider’s local privacy policy. In practice, nodes can choose to keep privacy-sensitive requests local, offloading only non-sensitive requests to the network according to their policy.

---

> > ### Comment · Reviewer_spMp · 2025-11-26
> >
> > Thank you for your detailed rebuttal and the new experiments. Your responses have clarified several key points and are much appreciated.
> >
> > I am largely convinced by your new experiment on network latency and your clarification regarding the adaptive duel rate. These have addressed some of my initial concerns about the system's robustness and overhead.
> >
> > However, my primary reservations still revolve around the gap between this idealized framework and its real-world viability. While the latency experiment is a good first step, the system's feasibility at a large scale remains unproven, especially concerning the unaddressed overhead of a true blockchain ledger. More fundamentally, the proposed solutions for critical issues like user data privacy and unaddressed "rich-get-richer" appear to sidestep the core challenges, potentially limiting the system's practical utility.
> >
> > Given your constructive response and the novelty of the research direction, I have raised my score to marginally acceptance. I am raising the score to acknowledge the paper's value as a research concept. However, my confidence remains neutral, as I believe the path to a practical and robust real-world deployment is still an open and critical question.

---

> > > ### Author Response · Authors · 2025-12-03
> > > **Response to Reviewer spMp**
> > >
> > > Thank you for your valuable feedback! We are glad that the additional experiments and clarifications addressed several of your initial concerns. Below, we provide further explanations on the remaining points.
> > >
> > > ## 1. System scalability and overhead of the blockchain ledger
> > >
> > > We would like to clarify that the core of our system lies in the PoS-based scheduling and the Duel-and-Judge mechanism: the latter continually adjusts credit based on observed service quality, while the former uses the resulting credit distribution to guide executor selection, both of which **operate independently** of the underlying ledger implementation. The blockchain ledger only serves as an existing and well-tested primitive that **provides tamper-resistant credit accounting in a fully decentralized environment**. Modern production blockchains have already demonstrated their ability to support large, globally distributed networks with significant throughput, as summarized in $\color{blue}{\text{Appendix G, Pg. 18}}$. As such, the ledger component is **not expected to be a limiting factor** for system scalability or overhead in our setting.
> > >
> > > ## 2. User data privacy
> > >
> > > We emphasize that our system **does not introduce any additional exposure of user data** beyond what service providers already permit under their existing operational policies. A central design goal of WWW.Serve is to support an open, competitive market of LLM services while **fully preserving service provider autonomy**: they determine when to participate, under what policies, and with what resources. Consequently, whether a user request may be forwarded to other nodes is entirely governed by each provider’s pre-established policies and remains **independent of their participation in WWW.Serve**. The framework therefore does not weaken user data privacy; it simply enables voluntary collaboration without altering the provider-level guarantees already in place.
> > >
> > > ## 3. Rich-get-richer dynamic
> > >
> > > In WWW.Serve, high-quality nodes receive positive net payoff while lower-quality nodes face penalties. However, a node’s credit trajectory is determined by the balance between **providing service and consuming resources**: nodes spend credits when offloading their own queries and earn credits by serving delegated requests. This creates a natural self-regulating feedback loop. We acknowledge that in **extreme scenarios**, where a node consistently produces high-quality responses and rarely offloads, some degree of rich-get-richer dynamics may emerge. However, such behavior reflects **genuine value creation and incentive alignment**: WWW.Serve is designed to encourage high-quality contributions. While such concentration would indeed be problematic if driven by malicious behavior, in the scenario above it arises precisely because the node is a net contributor to the system. Nonetheless, exploring complementary mechanisms, such as adaptive credit decay, to guard against pathological cases could further strengthen the system’s practical utility and remains an important direction for future work.

---

### Official Review · Reviewer_dWSd · 2025-11-06

**Soundness:** 3
**Presentation:** 3
**Contribution:** 3
**Rating:** 6
**Confidence:** 3

**Summary:**

The paper introduces WWW.Serve, a fully decentralized framework for collaborative LLM serving that aims to overcome the scalability and privacy limitations of centralized services. It does this through 3 mechanisms: a blockchain-inspired credit system for trustless request delegation, a gossip-driven protocol for dynamic peer synchronization, and a duel-and-judge mechanism for robust contributor evaluation. Empirical results show improved SLO attainment, by up to 1.5× and reducing latency by 27.6%—while maintaining robustness under dynamic participation and preserving provider privacy.

**Strengths:**

1. The system provides a lot of flexibility to service providers allowing them to design their own scheduling and load balancing policies for their infrastructure. This will incentivize more providers to join the ecosystem while obscuring the details of the backend from the users - who will have the same experience as a centralized service where they upload their queries and receive the response.

2. At the same time the credit-based transaction system and the duel-and-judge mechanism ensures that nodes cannot misreport their results or get away with producing quick but poor-quality responses. The authors show through both theoretical analysis and experiments that high quality nodes will accumulate stake share while low-quality node will be gradually phased out of the system.

**Weaknesses:**

1. The comparison with related works appears incomplete. I feel that for the list works (Kozgunovetal.,2024; Xianetal.,2024; Chenetal.,2025; Mia & Amini,2025) that explore secure, decentralized learning and inference frameworks, there should be an explicit statement on how well these works answer the 3 questions raised in the introduction, rather than a generic remark like, "overall, existing systems remain insufficient..."

2. It is not clear to me if the user has any say in which service provider manages their request. If they do not have a say in that then that is a drawback since many users may want to restrict their requests to certain service providers - especially due to privacy concerns and also due to variation in the scheduling policies of different providers.

3. The theoretical analysis only considers response quality and does not model the effect of high demand increasing the latency at the high performing nodes.

**Questions:**

1. Won't there be staleness in the information in the ledger since validation by peers will take time? And if there is staleness, won't that lead to incorrect scheduling?

2. Won't inference requests be blocked by requests for judging another node's response, thereby increasing the overall latency?

3. Can you provide an explanation/intuition for why the decentralized approach outperforms even the centralized one in some cases in Fig. 4?

4. The bars in Fig 5 are too thin, and it is difficult to understand what is going on in the plots. Please find an alternate way of representing those results.

5. Why does the Qwen 0.6B get the largest total number of requests in Fig 6?

6. What does 100% offloading in Section 6.4 mean? If all nodes offload everything, where does a request go? Also more generally, won't high offloading rates lead to requests being bounced around a lot and getting delayed because of that?

---

> ### Author Response · Authors · 2025-11-25
> **Response to Reviewer dWSd (part 1)**
>
> Thank you very much for your thoughtful review and constructive suggestions. We are glad that the reviewer found our **system design flexible and effective**. We have tried to address your questions carefully. We hope the reviewer will consider raising your score in light of our response.
>
> ## W1: How well previous works answer the 3 questions raised in the introduction?
>
> We have **revised the Abstract, Introduction, and Related Work** $\color{blue}{\text{(Section 1-2, Pg. 1-3)}}$ to clarify our motivation, highlight gaps in prior work, and demonstrate how our approach advances the field. In particular, we clarify the limitations of existing work in addressing the three key questions raised in the Introduction:
> 1. **Q1**: How can the system enable trustworthy market-driven trade of computational capacity, i.e., implement reliable request scheduling among anonymous participants without central coordinators?
> 2. **Q2**: How can we incentivize participants to provide high-quality services, thereby improving overall user experience?
> 3. **Q3**: How can the system remain robust under highly dynamic and unpredictable resource availability?
>
> Previous related works:
> 1. **C-LLM** (Xian et al., 2024) focus on improving LLM output trustworthiness through blockchain-based provenance, but they do not provide decentralized request routing (**Q1**) or mechanisms for adapting to highly dynamic resource availability (**Q3**).
> 2. **Linguachain** (Kozgunov et al., 2024), **Flock** (Chen et al., 2025), and **FedShield-LLM** (Mia & Amini, 2025) address decentralized training rather than online inference, so they fall outside the scope of our comparison regarding **Q1-Q3**.
> 3. **Petals** (Borzunov et al., 2023) supports collaborative inference but only for a single fixed model; it lacks a decentralized scheduling mechanism for multi-model routing (**Q1**) and provides limited support for rapidly changing volunteer resources (**Q3**).
> 4. **DeServe** (Wu et al., 2025) offers partially decentralized resource publication but still relies on centralized task matching, leaving **Q1** unresolved, and it does not include decentralized mechanisms for online quality assessment (**Q2**).
> 5. **GenTorrent** (Fang et al., 2025) depends on trusted organizations to prevent malicious behavior, which is incompatible with a fully decentralized and anonymous setting and therefore cannot address **Q1-Q3**.
>
> In contrast, our framework explicitly tackles all three questions through **PoS-based decentralized scheduling (Q1)**,  a **duel-and-judge mechanism for decentralized service-quality assurance (Q2)**, and a **gossip-driven resource-state synchronization protocol (Q3)**.
>
> ## W2: Whether the user has any say in which service provider manages their request?
>
> Our framework gives each service provider (node) **full control over how user-submitted requests are handled**. Based entirely on its own policy, a node may choose to execute requests locally or offload them to the network. Consequently, whether a user’s request remains local is determined by the **agreement or configuration between the user and the service provider** they choose to interact with. The decentralized framework only applies to requests that a provider explicitly opts to offload. Thus, users can effectively restrict execution to trusted providers when desired, without being forced into network-wide routing.
>
> ## W3: The theoretical analysis only considers response quality and does not model the effect of high demand increasing the latency at the high performing nodes.
>
> We agree that our theoretical analysis mainly focuses on the **relationship between node quality and accumulated credit**: high-performing nodes steadily accumulate credit, while low-quality nodes see reduced exposure and are gradually phased out. This establishes a **trust-aligned credit system**, which justifies PoS-based scheduling in selecting trustworthy executors from anonymous nodes, and ensures that overall LLM service quality is maintained.
>
> Importantly, we would like to clarify that all nodes in our system have **full control over whether to accept delegated requests from other nodes**. They can also adjust the amount of staked credits to influence their probability of being scheduled under our PoS design. Rational high-performing nodes will only accept additional delegated requests when their resources are available, and refrain from overcommitting when their capacity is limited. Therefore, our system can prevent severe latency increases at high-performing nodes.
>
> We have also **formalized the game-theoretic analysis** in $\color{blue}{\text{Section 5, Pg. 7-8}}$ with theorem statements to provide a clearer and more rigorous presentation.

---

> ### Author Response · Authors · 2025-11-25
> **Response to Reviewer dWSd (part 2)**
>
> ## Q1: Won't there be staleness in the information in the ledger? Won't that lead to incorrect scheduling?
>
> We agree that the ledger synchronization among peers may cause minor staleness. However, this has **negligible impact on scheduling correctness**. The ledger records staked credit as a slowly evolving consensus state, rather than a real-time metric. Our PoS scheduler only relies on the latest confirmed snapshot, which changes at a much slower rate than individual request cycles. Hence, temporary inconsistency merely introduces **minor randomness rather than bias**. Moreover, the system ensures bounded synchronization latency through periodic gossip, guaranteeing eventual consistency.
>
> ## Q2: Won't inference requests be blocked by requests for judging another node's response, thereby increasing the overall latency?
>
> We would like to clarify that, in our design, **a judging task is essentially an inference request** processed by the LLM servers and subject to the same batching and scheduling mechanisms. Therefore, judging and user requests are jointly scheduled, allowing GPU-level batching and efficient parallel execution. This design ensures that judgment tasks **do not increase end-to-end latency beyond normal inference variance**.
>
> We also provided an **ablation study on the duel-and-judge mechanism** in $\color{blue}{\text{Appendix F, Pg. 18}}$, quantifying its overhead through both theoretical analysis and empirical evaluation of latency and SLO attainment under different duel rates.
>
> ## Q3: Why does the decentralized approach outperform even the centralized one in some cases in Fig. 4?
>
> In our experiments, each node autonomously decides whether to accept or redirect a request based on its local workload, latency feedback, and scheduling policy. This flexible, self-adaptive behavior enables nodes to **react immediately to transient load fluctuations** without waiting for global coordination. In contrast, centralized scheduling **mainly optimizes global load balance**, which can be less responsive under rapidly changing or heterogeneous workloads. As a result, decentralized scheduling occasionally achieves better global SLO and lower average latency.
>
> ## Q4: The bars in Fig 5 are too thin, and it is difficult to understand what is going on in the plots.
>
> We would like to clarify that the main purpose of Fig. 5 is to illustrate how our system adapts to dynamic resource participation. **The black line shows the windowed average request latency**, which clearly rises or drops in response to node join/leave events, demonstrating the network’s resilience to changes in available resources. The colored bars indicate which nodes handled each request and serve as supplementary information; they are **not essential for interpreting the overall latency trends**.
>
> ## Q5: Why does the Qwen 0.6B get the largest total number of requests in Fig 6?
>
> The total number of duel-and-judge requests received by the 0.6B model is higher primarily due to a **time-budget effect rather than a scheduling-preference effect**. Our experiment fixes the evaluation window to 90 minutes. Because the 0.6B model generates responses significantly faster than the 4B and 8B models, it can simply participate in a larger number of duel rounds within the same time budget. This increases its raw request count, even though its probability of being scheduled at each round is strictly proportional to its staked credit, which is much lower than that of the more powerful (8B, 4B) models. From the perspective of the mechanism being evaluated, the more meaningful indicator is the **credit trajectory and win rate**. These metrics clearly show the expected ordering: 8B > 4B > 0.6B > random, demonstrating the effectiveness of our duel-and-judge mechanism. We have updated this subsection ($\color{blue}{\text{Subsection 6.3, Pg. 10}}$) to clarify the presentation of the experiments.
>
> ## Q6: What does 100% offloading in Section 6.4 mean? Where does a request go? Won't high offloading rates lead to requests being bounced around a lot and getting delayed?
>
> In Section 6.4, “100% offloading” denotes **the fraction of requests that the node’s policy has already marked for offloading that are actually redirected**. It does not mean the node offloads all incoming requests. This offloading percentage is a **controlled experimental knob** we vary to quantify the effect of more or less aggressive redirection. In real deployments, such an explicit “offloading frequency” does not exist. Offloading is entirely determined by the node’s policy and local load estimation.
>
> Actually, offloaded requests do not bounce arbitrarily among nodes. Offloading requires an explicit **acceptance from the receiving peer**, which may accept requests when it is sufficiently idle. As a result, requests are forwarded mainly to **less-loaded nodes rather than circulating**, keeping latency lower than simply queuing locally under overload.

---

### Author Response · Authors · 2025-11-25
**Response to All Reviewers**

We thank all the reviewers **[R1 (dWSd), R2 (spMp), R3 (SRSf), R4 (DmSJ)]** for their thoughtful and highly supportive feedback! We were glad that the reviewers found the problem **significant and timely [R2, R3, R4]**, our system design **comprehensive and flexible [R1, R2, R3]**, and the **breadth of our evaluation [R1, R2, R4]**. Reviewers further recognized the **originality of our framework [R4]**, and believed that it **could have a transformative impact on the field [R2]**.

We have updated the paper to incorporate constructive suggestions, as shown in the revision. We summarize the major changes:

1. **[R1(W1), R4(W1)] Revised Abstract, Introduction, and Related Work** to clarify our motivation, highlight gaps in prior work, and show how our approach advances the field. ($\color{blue}{\text{Section 1-2, Pg. 1-3}}$)
2. **[R1(W3), R2(W3, Q2), R4(W2)] Formalized game-theoretic analysis** with theorem statements and proofs. ($\color{blue}{\text{Section 5, Pg. 7-8}}$)
3. **[R4(Q2)] Clarified terminology** by defining key concepts referenced throughout the paper. ($\color{blue}{\text{Appendix D, Pg.15-16}}$)
4. **[R4(Q4-6)] Supplementary system details and new figures** illustrating the gossip-driven protocol and the end-to-end workflow of a single user request. ($\color{blue}{\text{Appendix E, Pg. 16-17}}$)
5. **[R1(Q2), R2(W2, Q1), R3(Q2)] Ablation study on the duel-and-judge mechanism** quantifying its overhead through both theoretical analysis and empirical evaluation of latency and SLO attainment under different duel rates. ($\color{blue}{\text{Appendix F, Pg. 18}}$)
6. **[R2(W1, Q4), R3(W1, Q1)] Summary of throughput and latency** of production blockchain systems for reference. ($\color{blue}{\text{Appendix G, Pg. 18}}$)

---

> ### Author Response · Authors · 2025-11-29
> **Response to All Reviewers**
>
> In addition, although not included in the revision, we conducted further experiments during the rebuttal phase to directly address reviewer concerns, including:
>
> 1. **[R2(W1, Q4)] System performance under varying network latency**. By injecting synthetic delays, we show that WWW.Serve remains stable when the network latency is within 300 ms, while performance gradually degrades as latency increases beyond this range, as expected.
>
>     | Synthetic Delay (s) | Avg Latency (s) | Avg Throughput (token/s) | p95 Latency (s) | p99 Latency (s)  |
>     |------------------------|-----------------------|-----------------------------------|----------------------|-----------------------|
>     | 0                          | 75.1897             | 87.86                                | 120.9926           | 129.2882            |
>     | 50                        | 73.9624             | 87.86                                | 114.5052           | 117.7532            |
>     | 100                      | 74.5123             | 86.96                                | 116.4965           | 124.5898            |
>     | 300                      | 76.2464             | 86.25                                | 118.4419           | 132.2803            |
>     | 500                      | 77.9760             | 84.50                                | 117.2979           | 129.9174            |
>     | 1000                    | 79.1607             | 80.14                                | 117.3263           | 122.5198            |
>
> 2. **[R3(W2)] Comparison with Petals**. Under identical hardware and model settings, WWW.Serve exhibits consistently lower inference latency and higher throughput across multiple generation lengths.
>
>     | System | Gen length | Latency (s) | Throughput (token/s) |
>     |------------|----------------|-----------------|-----------------------------|
>     | Petals   | 128             |  17.8931     | 7.15                          |
>     | WWW.Serve   | 128   |  **2.8366**       | **45.12**                         |
>     | Petals   | 1024           |  142.0986   |  7.21                          |
>     | WWW.Serve  | 1024  |  **21.1692**     |   **48.37**                       |
>     | Petals   | 4096           |  542.2454    | 7.55                          |
>     | WWW.Serve  | 4096  |  **76.4542**     |  **48.60**                        |

---

### Author Response · Authors · 2025-11-29
**Overview for the Newly Assigned Area Chair**

Thank you for taking on the assessment of our submission given the exceptional review circumstances this year. We appreciate your additional effort. To facilitate an efficient review, we provide a brief summary of the discussion history.

Our paper initially received scores of **2, 6, 4, 6** with corresponding confidences of **2, 2, 3, 3**. During the discussion phase, **Reviewer spMp** updated their score from **4 to 6**, resulting in an updated set of scores: **2, 6, 6, 6**. This change was recorded at **26 Nov, 8:38AM EST**, **prior to** the OpenReview malfunction that occurred around 27 Nov, 10AM EST. At that time, the other three reviewers had not yet responded to our rebuttal.

---

### Meta-Review · Area_Chair_GcqC · 2026-01-07

**Summary:**

Across reviewers, the primary concern is a mismatch between the paper’s claims (large-scale, decentralized, practical serving) and the level of validation provided. The evaluation is conducted at very small scale (4–8 nodes) under idealized network conditions, relying on a shared ledger rather than a full blockchain, leaving scalability, gossip convergence, ledger throughput, and communication overheads unaddressed. Core mechanisms such as the duel-and-judge process and the credit ledger incur non-trivial overheads that are neither rigorously quantified nor justified. The theoretical analysis is considered overly idealized, assuming static node quality and rational behavior, omitting demand-induced latency, and lacking formal assumptions, theoremized results, and convergence guarantees. Reviewers also raise concerns about long-term dynamics, noting that the proposed credit evolution may lead to re-centralization via “rich-get-richer” effects. In addition, privacy claims are viewed as incomplete, as user data protection is not addressed. Finally, the paper’s positioning is weakened by insufficient comparison with closely related decentralized serving systems and by unclear motivation, definitions, and presentation in several sections. These issues undermine the paper’s claims and practical relevance, leading to an overall recommendation of rejection.

**Reviewer Concerns:**

The author's rebuttal explicitly addresses the reviewers' concerns regarding the system's robustness and computational overhead. The author clarifies key concepts and analyses through formal means such as theorem statements, proofs, and definitions, while optimizing the writing to make the motivation clearer. The authors clarify key concepts and analyses through formal methods such as theorem statements, proofs, and definitions, while optimizing the writing to enhance clarity in motivation and workflow. However, reviewers expressed concerns about whether the proposed framework could operate in the real world, particularly regarding its feasibility in larger-scale systems. Additionally, there is a lack of direct solutions for user data privacy and the “rich-get-richer” problem. The authors’ rebuttal did not directly provide verifiable experiments to refute these core issues.

**Reviewer Scores:**

Reviewer spMp explicitly stated that the rating would be raised to 6. I expect the final rating to be as follows:
- Reviewer dWSd: 6
- Reviewer spMp: 6
- Reviewer SRSf: 6
- Reviewer DmSJ: 2

---

### Decision · Program_Chairs · 2026-01-26

Reject